# The Diurnal Blood Metabolome and Effects of Vitamin D Supplementation: A Randomised Crossover Trial in Postmenopausal Women

**DOI:** 10.3390/ijms23179748

**Published:** 2022-08-28

**Authors:** Rasmus Espersen, Banny Silva Barbosa Correia, Lars Rejnmark, Hanne Christine Bertram

**Affiliations:** 1Department of Endocrinology and Internal Medicine, Aarhus University Hospital, Palle Juul-Jensens Boulevard 99, 8200 Aarhus, Denmark; 2Department of Clinical Medicine, Aarhus University, Palle Juul-Jensens Boulevard 82, 8200 Aarhus, Denmark; 3Department of Food Science, Aarhus University, Agro Food Park 48, 8200 Aarhus, Denmark

**Keywords:** NMR metabolomics, vitamin D metabolism, rhythmicity, blood metabolites, diurnal rhythm

## Abstract

A way to maintain an adequate vitamin D status is through supplementation. Demonstration of blood-metabolome rhythmicity of vitamin D_3_ post-dosing effects is lacking in the pharmaco-metabonomics area. Thus, the overall aim of this study was to investigate the diurnal changes in the blood metabolome and how these are affected by vitamin D_3_ supplementation. The study was conducted as a crossover study, and the treatment included 200 µg (8000 IU) of vitamin D_3_ as compared with placebo with a washout period of at least 10 days. The participants were postmenopausal women aged 60–80 years (N = 29) with vitamin D insufficiency (serum 25-hydroxyvitamin D < 50 nmol/L) but otherwise healthy. During the intervention day, blood samples were taken at 0 h, 2 h, 4 h, 6 h, 8 h, 10 h, 12 h, and 24 h, and plasma was analysed by proton nuclear magnetic resonance (NMR) spectroscopy as a metabolomics approach. In general, diurnal effects were identified for the majority of the 20 quantified metabolites, and hierarchical cluster analysis revealed a change in the overall plasma metabolome around 12 AM (6 h after intervention), suggesting that the diurnal rhythm is reflected in two diurnal plasma metabolomes; a morning metabolome (8–12 AM) and an afternoon/evening metabolome (2–8 PM). Overall, the effect of vitamin D supplementation on the blood metabolome was minor, with no effect on the diurnal rhythm. However, a significant effect of the vitamin D supplementation on plasma acetone levels was identified. Collectively, our findings reveal an influence of diurnal rhythm on the plasma metabolome, while vitamin D supplementation appears to have minor influence on fluctuations in the plasma metabolome.

## 1. Introduction

The majority of total body vitamin D is derived from synthesis in the skin in response to sun exposure [1,2]. However, in countries north of the 37° N latitude, there is no endogenous production of vitamin D during the wintertime [2,3,4]. As a consequence, serum 25-hydroxy vitamin D (25(OH)D), the biochemical marker of vitamin D nutritional status, exhibits a seasonal rhythm, with highest values at the end of summer and lowest at the end of winter [5]. Calcitriol, vitamin D-binding protein, and all major hormones and minerals related to the calcium homeostasis display a diurnal rhythm [6]. Recently, a case report demonstrated that 25(OH)D displays a diurnal rhythm as well [7].

Metabolomics is an advanced, post-genomic approach that is used for the chemical profiling and/or quantification of as many metabolites as possible in order to constitute the so-called metabolome [8]. The metabolome provides an expression of an individual’s metabolic state and may predict their response to different interventions [9]. The investigation of the metabolite profile of blood and other biofluids has been assessed by different analytical methods, e.g., proton nuclear magnetic resonance (NMR) spectroscopy, which allows analyses to be conducted with relatively easy sample preparation and the possibility of a non-selective and simultaneous qualitative and quantitative evaluation of a wide range of metabolites [10,11]. NMR-based metabolomics has been widely applied in the analysis of blood samples by research groups worldwide [12,13].

Blood is extensively studied as it reflects the metabolites circulating in the body and thereby the metabolites that the cells within our body are exposed to as well [14]. It has been demonstrated that the blood metabolome exerts diurnal rhythmicity [10,14,15,16,17,18], and that certain conditions and interventions affect this diurnal rhythm [19,20]. Furthermore, the possibility of predicting post-dose drug effects from baseline metabolic profiles has been proposed (pharmaco-metabonomics) in several studies [9,21,22].

However, knowledge about the acute effect of a single, high dose of vitamin D_3_ on the metabolome is sparse. To the best of our knowledge, only one study has investigated the acute effect of a single dose of vitamin D_3_ on the metabolome. The VITdAL-ICU trial [23] randomised 428 critically ill participants with vitamin D insufficiency to receive either a single dose of 13,500 µg (540,000 IU) vitamin D_3_ or a placebo. In a post-hoc metabolomics study, the authors found significant changes in the metabolome over time relative to the absolute increase in serum 25(OH)D levels from day 0 to day 3. Increases in sphingomyelins, plasmalogens, lysoplasmalogens and lysophospholipids (i.e., metabolites involved in endothelial protection and innate immunity), as well as decreases in phosphatidylethanolamines, amino acid metabolites, and acylcarnitines (i.e., metabolites involved in mitochondrial function and fatty acid beta-oxidation) were found. Others have found an association between lipid metabolism and fatty acid oxidation and vitamin D as well. Shirvani et al. [24] reported an increase in metabolites mostly involved in the oxidation of branched-chain fatty acids after 24 weeks with supplementation of either 15 µg, 100 µg, or 250 µg of vitamin D_3_ per day, and vitamin D status was associated with lipid metabolism or metabolites of lipid-compound origin in the ATBC Cancer Prevention Study [25] and the Hong Kong Osteoporosis Study [26], respectively. However, these metabolic evaluations mainly regarded investigations into lipids lacking polar metabolites. Recently, it was discovered that a daily supplement of 70 µg (2800 IU) vitamin D_3_ over 12 weeks increased circulating levels of carnitine, choline, and urea (i.e., muscle-related metabolites negatively correlated with muscle health and physical performance) in postmenopausal women with vitamin D insufficiency [27]. Combined with clinical findings that reported negative effects of vitamin D on muscle strength and physical performance [28], this suggested a detrimental effect of moderately high daily doses of vitamin D supplementation on skeletal muscle. However, the daily doses of vitamin D supplementation in the above-mentioned studies were all very high, and in the VITdAL-ICU trial, a mega-dose of 13,500 µg was studied, i.e., multiple times above the 100 µg (4000 IU) per day which often is considered the upper limit of a safe intake [29]. Thus, little is known about the effect of a moderate vitamin D dose on the metabolome. Therefore, the aim of this study was to investigate the diurnal blood metabolome in response to a single, high dose of vitamin D_3_. Our null hypothesis was that the metabolome was unaffected 24 h after vitamin D_3_ intake, i.e., generating parallel time-concentration curves with no difference between curves.

## 2. Results

### 2.1. Demography

A total of 29 women with a mean age of 69 years were included in the analyses. Of these, 15 were given vitamin D_3_ as an intervention on the first day. Analyses did not suggest a period effect or a carry-over effect. Table 1 shows the characteristics of the participants. Mean 25(OH)D levels were 39 nmol/L at screening, 45 nmol/L at baseline on Day 1, and 50 nmol/L at baseline on Day 2.

### 2.2. Diurnal Effects

The obtained NMR spectra were analysed using the In Vitro Diagnostics research (IVDr) tool which, for plasma samples, registers 41 metabolites from a database. However, only 22 plasma metabolites registered signal intensities adequate for accurate identification and quantitative evaluation in all of the participants’ plasma (Appendix A). For further investigations of an effect of vitamin D supplementation, we excluded EDTA bound with endogenous plasma calcium and potassium, leaving 20 metabolites for the investigative statistical analyses.

The metabolites showed different fluctuations over time during the 24 h sampling period (Appendix A). Figure 1 shows selected metabolites representing the different fluctuation patterns observed. Plasma glucose concentrations varied considerably. Initially, a decrease in glucose concentration was observed after the intervention and morning meal (from 8 AM to 10 AM) and subsequently, it increased substantially after the second meal (from 12 PM to 2 PM), where a second drop was seen. Amino acids, including alanine, tyrosine, and isoleucine concentrations also displayed fluctuations characterised by an initial decrease (from 8 AM to 12 PM) followed by a steady increase. Lactic acid displayed less systematic fluctuation in concentration during the 24 h period. Acetone showed an increase that peaked around 6–8 PM, corresponding to the timing of the evening meal, after which it decreased again.

In order to elucidate which metabolites showed the most variation as a function of the experimental design (factors including time and vitamin D supplementation), a univariate one-way ANOVA analysis with Fisher’s LSD test was employed as an explorative approach. ANOVA is commonly used to compare the means or medians of one variable across two or more groups [30]. In this case, ANOVA was used across 16 groups corresponding to the 8 time-points and 2 treatments (vitamin D versus placebo). In Figure 2, results from the exploratory, one-way ANOVA are shown. The concentrations of 15 metabolites differed between the 16 groups, as shown in Figure 2A. The metabolites with the highest (−)log10 *p*-value display the most variation across the 16 groups, and the metabolites isoleucine, phenylalanine, glucose, tyrosine, and leucine were the most significant metabolites. Furthermore, as shown in Figure 2B by heatmap clustering and in the Fischer LSD analysis reported in Appendix A, the 15 metabolites showed a clear separation into two main clusters. Cluster 1 represents time-points between 8 AM and 12 PM, including the two fasting time-points (0 h and 24 h), and Cluster 2 represents time-points between 2 PM and 8 PM. The plasma metabolites generally showed higher concentrations in Cluster 2 than in Cluster 1, reflecting a diurnal rhythm of the metabolites.

### 2.3. Vitamin D_3_ vs. Placebo

In general, no pronounced effects of vitamin D supplementation were indicated by the explorative one-way ANOVA analysis, while time effects dominated the identified variations. However, when the relative abundance was compared using heatmaps, a slight difference occurred between vitamin D supplementation and placebo 12 h after the intervention (8 PM), where valine showed a significant decrease after vitamin D supplementation in comparison with the placebo, as seen in Appendix A.

For each of the 20 metabolites, the test for parallel mean time-concentration curves for the two interventions showed no statistically significant difference (*p* > 0.15) (Table 2). When comparing the baseline adjusted mean curves, there was no difference between the two interventions for acetic acid, alanine, citric acid, creatine, creatinine, dimethyl-sulfone, formic acid, glucose, glycine, histidine, isoleucine, lactic acid, leucine, phenylalanine, pyruvic acid, succinic acid, trimethyl-amine-N-oxide, tyrosine, or valine. However, acetone was 0.003 mmol/L (95% CI: 0.002; 0.005) higher during the 24 h after the administration of vitamin D_3_ compared to that of placebo (*p* < 0.001) (Table 2).

## 3. Discussion

In this randomised crossover study of the metabolome 24 h post-dosing with 200 µg vitamin D_3_ or placebo, we found no evidence that vitamin D_3_ altered the diurnal rhythm of the plasma metabolome. Among 20 quantified metabolites, we found that vitamin D_3_ increased the concentration of acetone, while the other 19 metabolites remained unaltered. 

Together with β-hydroxybutyrate and acetoacetate, acetone is a ketone body, and ketone bodies represent a set of fuel molecules serving as an alternative energy source to glucose. Ketone bodies are mainly produced by the liver from fatty acids, often during periods of fasting and prolonged or intense physical activity. Ketogenesis is intensified under conditions characterised by the insufficient or inaccessible availability of glucose, and higher acetone levels have also been reported in individuals with diabetes [31]. In fact, diurnal fluctuations in acetone were initially investigated in connection with the establishment of insulin therapy [32]. In the present study, the effect on acetone could indicate an effect of vitamin D on the beta-oxidation of fatty acids and thereby the formation and accumulation of ketone bodies. Recently, it was discovered that lactone-vitamin D_3_, a vitamin D_3_ metabolite, impairs fatty acid beta-oxidation in vitro [33]. In the VITdAL-ICU trial, the authors found decreased concentrations of metabolites associated with beta-oxidation after administration of vitamin D_3_ [23]. 

The finding that valine, a branched-chain amino acid, decreased 12 h after administration of vitamin D could point to the intervention influencing the diurnal rhythm of this metabolite [34]. Some studies have shown higher levels of branched-chain amino acids in the plasma of individuals with diabetes [35]. Dallmann et al. have correlated branched-chain amino acids in saliva with circadian effects showing variations over 24 h equal to, or up to 10-fold greater than, the differences considered diagnostic of diabetes [18]. Thus, data could indicate an effect of vitamin D on insulin sensitivity 12 h after administration. However, this remains a weak indication since only one branched-chain amino acid was affected and only at one time-point.

A major strength of this study is its crossover design, which reduces the risk of confounding due to interindividual variation. While a potential issue with crossover trials can be that of carry-over effects, no evidence of a carry-over effect was detected, suggesting that a washout period of at least 10 days was sufficient. Furthermore, the study was carried out during the wintertime when there is no or low endogenous vitamin D production. Another strength of this study is its homogenous population of Caucasian, postmenopausal women. However, this is also a limitation since our results cannot necessarily be generalised to men, premenopausal women, or women of non-Caucasian ethnicity.

The use of multiple testing and the risk for type-I errors is a limitation of this study. However, we used Bonferroni corrections to comply with this risk. Bonferroni corrections, on the other hand, are a conservative method, increasing the risk of type-2 errors.

NMR-based metabolomics has been documented as being an extremely reproducible analytic method [36,37], and in the present study, absolute plasma metabolite concentrations were obtained, which is a strength compared with metabolomic approaches based on mass spectrometry (MS). On the other hand, NMR spectroscopy is restricted in the number of metabolites that it can detect, which is a limitation as compared with MS-based approaches.

In conclusion, we confirmed the findings of other authors regarding a diurnal rhythm in the metabolome. This rhythm was not altered by the administration of 200 µg of vitamin D_3_. However, the concentration of acetone increased after the administration of vitamin D, suggesting an effect on beta-oxidation. Further studies are needed to further investigate these findings.

## 4. Materials and Methods

We enrolled 30 Caucasian, postmenopausal women with vitamin D insufficiency, i.e., 25(OH)D levels < 50 nmol/L, aged between 60 and 80 years. Exclusion criteria were daily use of calcium and/or vitamin D supplementation (>10 µg/day); current treatment with beta-blockers; overt cardiovascular disease; chronic kidney or liver disease; cancer; treatment for rheumatoid arthritis; treatment with oral corticosteroids or lithium; allergy or intolerance to milk, juice, or vitamin D supplementation; planned travel south of the 37° N latitude in the study period; and the use of sun beds.

Participants were recruited from the general background population by direct mailing using a list of randomly selected individuals living in the area of Aarhus, Denmark. In total, 8977 invitations were sent out. Of 774 respondents, 243 were eligible for a blood test. A total of 71 had a vitamin D insufficiency. Among these, 30 gave their consent to participate in the study, 1 of whom dropped out. A total of 29 participants were included in the present analyses (Figure 3). 

The study was conducted at Aarhus University Hospital, Denmark, at the latitude of 56° N during the winter and spring of 2019 and 2020 as part of a larger study investigating the influence of the food-matrix delivery system on the bioavailability of vitamin D_3_ (www.clinicaltrials.gov ID: 1107213017) (accessed on 8 April 2022). The study was performed in accordance with the Helsinki II declaration. Participation was preceded by written consent, and the protocol was approved by the ethical committee in the Central Denmark Region (1-10-72-130-17).

### 4.1. Study Design

The analyses were based on data from a randomised, multiple crossover study (with five treatment arms). Before inclusion, 25(OH)D levels were measured to allow for the inclusion of only women with vitamin D insufficiency. The current analyses included only two of the treatment arms, during which the metabolome was investigated in response to 500 mL of water, either with or without 200 µg (8000 IU) of vitamin D_3_ added. Treatment sequences were randomised using the sealed-envelope method, with each treatment being investigated on two separate days. If a participant received water with vitamin D_3_ on the first study day, water without vitamin D_3_ (placebo) was served on the second study day, and vice versa. The participants were blinded. A washout period of at least 10 days was interposed between the two study days. Before the implementation of the treatment arms in question, some of the participants had been allocated to other treatment arms (also including a single dose of 200 μg of vitamin D_3_), and accordingly, not all participants were strictly vitamin D insufficient during current measurements.

On each study day, a catheter was placed in an antecubital vein for blood sampling. After we obtained a fasting blood sample at 8 AM, the participants received the intervention together with a light breakfast meal composed of bread, jam, and coffee or tea. Blood samples were drawn 2 h, 4 h, 6 h, 8 h, 10 h, 12 h, and 24 h after the fasting blood sample (0 h). The participants had lunch after the 4 h blood sample, and supper after the 10 h blood sample. The 24 h blood sample was collected in an overnight fasting state (Figure 4).

### 4.2. Measurements

For screening and baseline status, serum 25(OH)D_2_ + D_3_ was measured immediately by tandem mass spectrometry and analysed by high-performance liquid chromatography with a coefficient of variation (CV) of 5.4% at 33 nmol/L and 10% at 113 nmol/L. Plasma creatinine was measured by absorption photometry and analysed using a chromogenic enzymatic reaction (Chemistry XPT, Siemens Healthcare Diagnostics, Inc., Erlangen, Germany) with a CV of 8.9% at 68 μmol/L (reference range: 45–90 μmol/L). 

Height and weight were measured without shoes and clothes except underwear. Height was measured to the nearest 0.1 cm using a wall-mounted stadiometer (seca, Hamburg, Germany). Weight was measured to the nearest 0.1 kg using an upright scale (seca, Hamburg, Germany). Body Mass Index (BMI) was calculated as weight divided by height squared (kg/m^2^). 

EDTA-plasma samples were frozen at −80 °C until analysed in a single batch for NMR analyses.

### 4.3. NMR Analyses

The plasma samples were thawed on ice and gently inverted a couple of times. For each sample, 350 µL of plasma buffer for IVDr (deuterium oxide phosphate buffer 0.10 M, pD = 7.4 containing 3-(trimethylsilyl)-propionic-2,2,3,3-d4 acid, sodium salt (TSP d4) 0.08%, and 0.04% of sodium azide) was added to an Eppendorf tube, and 350 µL of plasma was added. The tubes were mixed for 2 min at 350 rpm with a table mixer, and 600 µL of the solution was transferred to a 5 mm NMR tube.

NMR spectra were recorded at 310K on a 600 MHz Bruker Avance III spectrometer (Bruker BioSpin, Rheinstetten, Germany) equipped with a 5 mm ^1^H-optimised double-resonance, broad-banded probe (broadband inverse) and fitted with the Bruker SampleJet robot cooling system set to 5 °C. Proton NMR spectra were acquired using the Bruker IVDr methods. The instrument was calibrated before the analysis, and the automated methods were performed on each sample. For each sample, 4 experiments were completed in automation mode, amounting to a total of 25 min acquisition time per sample: 1D NOESY presaturation experiment (32 scans using a mixing time of 0.01 s and a relaxation delay of 4 s, 96K data points, a spectral width of 30 ppm, line-broadening of 0.3 Hz, zero-filled to 128K); a 1D−Carr−Purcell−Meiboom−Gill (CPMG) spin-echo experiment (32 scans, 72K data points, a spectral width of 20 ppm, and a spin-echo time of 300 μs, with a train of 128 refocusing pulses, line-broadening of 0.3 Hz, zero-filled to 128K); a 2D J-resolved experiment (2 scans, 40 t1 increments with 2 scans each, spectral width 16 ppm, line-broadening of 0.3 Hz in F2, Qsine weighting in F1 and F2, zero-filled to 256K in F1 and 16K in F2), and a 1D diffusion-edited spectrum (diffusion data was not used in the paper). ^1^H-^13^C HSQC experiments were performed on selected samples. 

After each spectral acquisition, the data were processed automatically using the automation routine. The line-broadening was set to 0.3 Hz; a zero-filling by a factor of 2 was used to produce 132K data points for processing, and finally, the first-order phase correction was set to 0.0. The obtained spectra were automatically phased using only zero-order phase correction, the TSP signal was calibrated to 0.0 Hz (0.0 ppm), and the spectral reference was recorded to be called into the next dataset acquisition. The concentration results were reported in mM to the IVDr metabolites list for plasma samples using Quantification Method Version Quant-PS 2.0.0 (Bruker BioSpin, Rheinstetten, Germany). The IVDr metabolites’ identities were confirmed by HSQC data.

All analyses were performed blinded.

### 4.4. Statistics

Two statistical approaches were applied in the present study. 

First, as an explorative investigation, a univariate method was employed based on one-way analysis of variance (ANOVA). Concentrations of plasma metabolites at different time-points after vitamin D_3_ supplementation or administration of the placebo were analysed using the MetaboAnalyst 5.0 platform (http://www.metaboanalyst.ca/faces/home.xhtml) (accessed on 8 April 2022) for testing significance of time and treatment without taking the order of time into account. Metabolite data were pareto-scaled (i.e., mean-centred and divided by the square root of the standard deviation of each variable), and their distribution was confirmed to be normal and homoscedastic. One-way ANOVA applied an adjusted *p*-value false-discovery rate (FDR) cut-off of 0.05, followed by post-hoc analysis using Fisher’s Least Significant Difference (LSD). Results were also viewed as combined in Hierarchical Clustering Heatmaps using Euclidean distance metrics and Ward’s clustering algorithm in ANOVA as the view option. 

Second, examinations of the concentrations of individual metabolites in response to time and treatment were performed. Period effects were investigated, comparing baseline values at Days 1 and 2, independently of whether vitamin D_3_ was provided or not, using the paired t-test. Using the Student’s t-test, carry-over effects were investigated by comparing differences in baseline values between Days 1 and 2 when vitamin D_3_ was provided on Day 1 with differences between baseline values on the 2 study days when the placebo was provided on Day 1.

Mean curves of time vs. concentrations of each metabolite were analysed using a mixed-effects regression model with intervention and time along with the interaction between them, order, carry-over, and day as fixed effects. Subject and day-within-subject were included in the analyses as random effects. Upon analysis, baseline adjustment was performed, subtracting baseline concentrations from each measurement for each metabolite. Model validation was performed by inspecting QQ plots for the standardised residuals and the plots of standardised residuals against the fitted values. 

Results are presented as estimated mean differences with 95% confidence intervals (CI).

A two-tailed *p*-value below 0.05 was considered statistically significant. However, to adjust for multiple testing, the Bonferroni corrected *p*-value, 0.05/20 = 0.0025, was considered to be statistically significant.

Data were collected and managed using REDCap electronic data-capture tools hosted at Aarhus University, Denmark [38,39], and they were analysed using Stata version 17 (StataCorp LLC, College Station, TX, USA) and IBM SPSS Statistics version 28 (IBM, Armonk, NY, USA). 

## Figures and Tables

**Figure 1 ijms-23-09748-f001:**
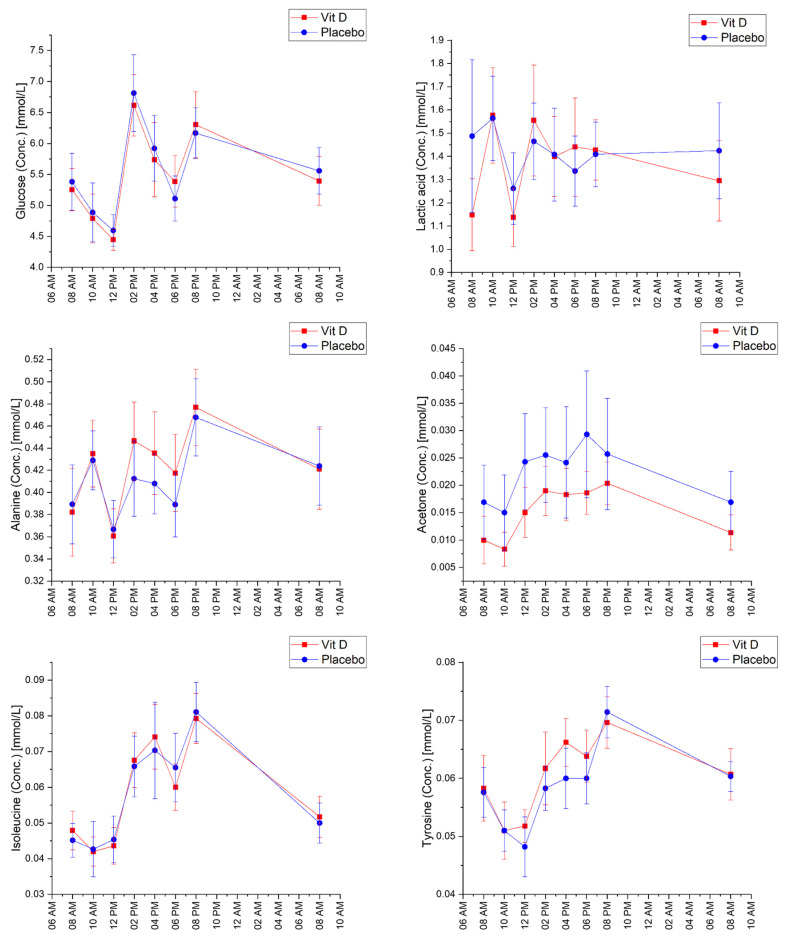
Concentrations in mM (Mean ± 95% CI) of plasma metabolites over time in participants after vitamin D_3_ supplementation and placebo.

**Figure 2 ijms-23-09748-f002:**
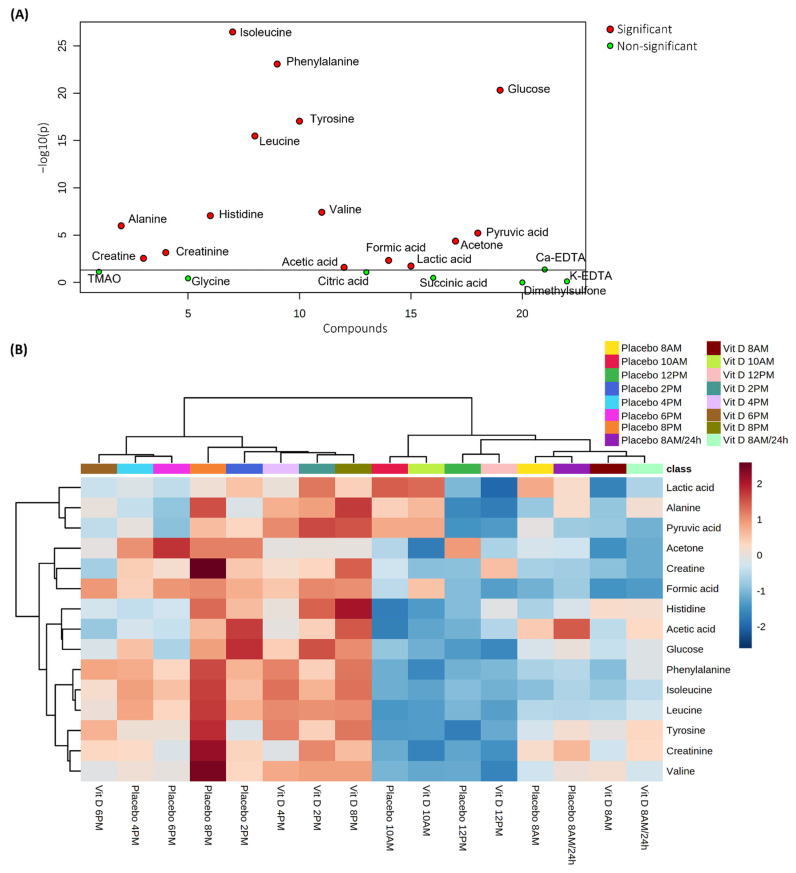
Variance analysis of plasma metabolite concentrations from different time-points after either vitamin D supplementation or placebo. (**A**) Scatterplot showing important features selected by ANOVA with a *p*-value threshold of 0.05. Red dots show significant metabolites, and green dots show non-significant metabolites. The line on the scatter plot indicates the (−)log10 *p*-value on the Y-axis. (**B**) Hierarchical Clustering Heatmap showing the distribution of the ANOVA-significant metabolites. Increased and decreased metabolite concentrations are given in red and blue, respectively. See Appendix A for the complete one-way ANOVA results.

**Figure 3 ijms-23-09748-f003:**
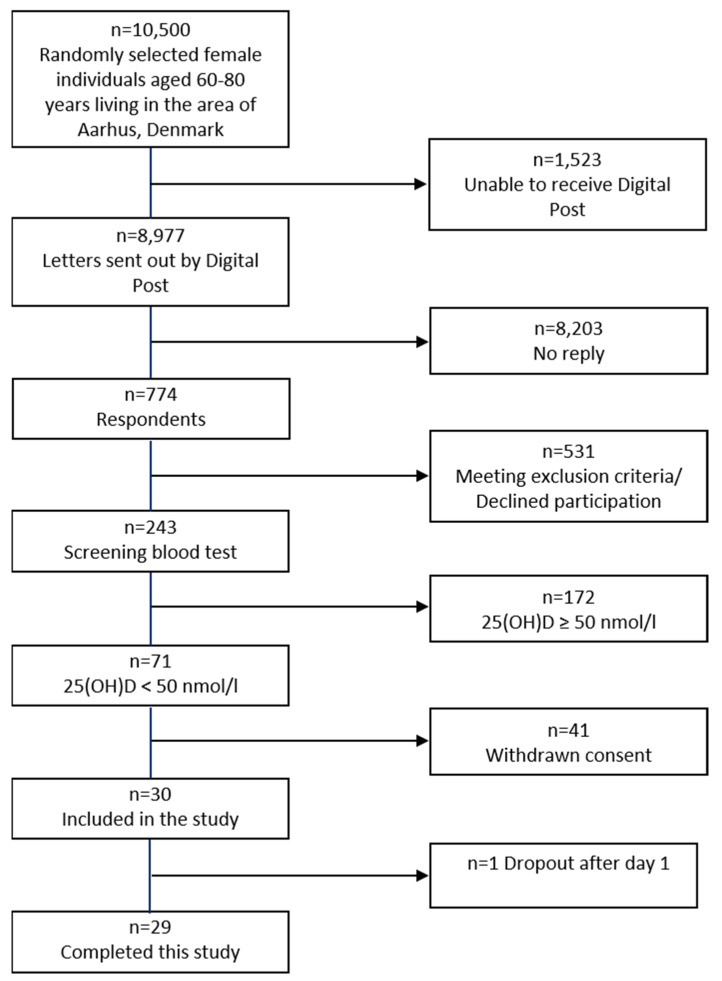
Flowchart of recruitment.

**Figure 4 ijms-23-09748-f004:**
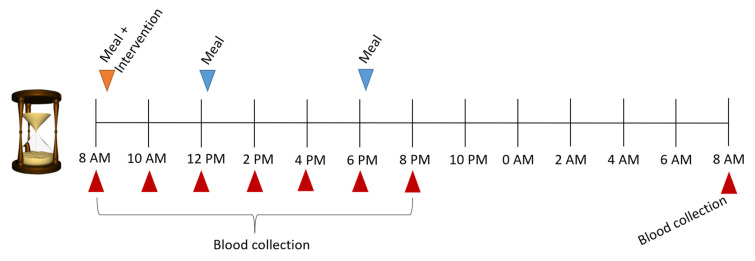
Study day timeline.

**Table 1 ijms-23-09748-t001:** Characteristics of included women.

		**N**	**Mean (SD)**	**Reference Interval**
Age, years		29	69 (4.11)	60–80
^a^ BMI, kg/m^2^		29	28 (6.93)	18.5–24.9
**Plasma**				
Screening	Creatinine, µmol/L	29	58 (10.54)	45–90
Screening	^b^ eGFR/1.73 m^2^, mL/min	29	88 (10.42)	>60
Screening	25-hydroxy vitamin D, nmol/L	29	39 (7.67)	>50
Day 1	25-hydroxy vitamin D, nmol/L	29	45 (11.36)	>50
Day 2	25-hydroxy vitamin D, nmol/L	29	50 (11.05)	>50

Abbreviations: ^a^ BMI: Body Mass Index; ^b^ eGFR: estimated glomerular filtration rate.

**Table 2 ijms-23-09748-t002:** Test for same development over time (i.e., parallel mean curves) and pairwise comparisons of mean baseline-adjusted metabolite concentrations after vitamin D supplementation and placebo.

Metabolite	Test of Same Time Effect *p*-Value	Mean Difference (95% CI) between Vitamin D and Placebo, mmol/L
Acetic acid	0.50	0.003 (0.000; 0.005), *p* = 0.02
Acetone	0.57	0.003 (0.002; 0.005), *p* < 0.001
Alanine	0.91	−0.002 (−0.017; 0.014), *p* = 0.84
Citric acid	0.98	−0.007 (−0.017; 0.003), *p* = 0.17
Creatine	0.28	0.000 (−0.005; 0.005), *p* = 0.96
Creatinine	0.29	−0.001 (−0.004; 0.002), *p* = 0.43
Dimethyl-sulfone	0.88	0.000 (−0.001; 0.000), *p* = 0.12
Formic acid	0.87	0.001 (−0.001; 0.004), *p* = 0.36
Glucose	0.91	−0.043 (−0.264; 0.179), *p* = 0.71
Glycine	0.93	−0.001 (−0.009; 0.008), *p* = 0.90
Histidine	0.96	−0.004 (−0.007; 0.000), *p* = 0.04
Isoleucine	0.62	0.001 (−0.003; 0.004), *p* = 0.69
Lactic acid	0.44	0.094 (0.006; 0.182), *p* = 0.04
Leucine	0.70	0.002 (−0.003; 0.006), *p* = 0.53
Phenylalanine	0.80	0.002 (−0.002; 0.006), *p* = 0.26
Pyruvic acid	0.76	0.005 (0.000; 0.010) *p* = 0.07
Succinic acid	0.67	0.000 (0.000;0.001), *p* = 0.27
Trimethyl-amine-N-oxide	0.68	0.008 (0.001; 0.014), *p* = 0.02
Tyrosine	0.42	0.000 (−0.002; 0.003) *p* = 0.89
Valine	0.16	−0.005 (−0.012; 0.002) *p* = 0.18

## Data Availability

The data that support the findings of this study are available on request from the corresponding author. The data are not publicly available due to privacy or ethical restrictions.

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
