# Peer review of "The Diurnal Blood Metabolome and Effects of Vitamin D Supplementation: A Randomised Crossover Trial in Postmenopausal Women"

_ijms, 2022, doi:10.3390/ijms23179748_

Round 1

Reviewer 1 Report

I am not an expert on the medical implications of the research, thus I restricted my review to the NMR and stats sections only. These are all excellent and of high quality, with appropriate reporting. I recommend accepting the paper in its current form.

Author Response

We appreciate the reviewer comment. Thank you for your acceptance recommendation.

Reviewer 2 Report

In the paper “The Diurnal Blood Metabolome and Effects of Vitamin D Supplementation: A Randomised Crossover Trial in Postmenopausal Women” authors have shown the effect of vitamin D supplementation on the blood metabolome on the diurnal rhythm. This manuscript required a major revision before considering for publication.

1.      What could be the potential source of acetone metabolism?

2.      Has acetone studied in metabolism in this type of study? Discuss with appropriate references.

3.      Is that really acetone or other ketone bodies?

4.      NMR spectra were recorded not conducted. Correct the sentence.

5.      How did you confirm pyruvate and acetone in 1D spectrum?

Author Response

REVIEWER 2

Comment: In the paper “The Diurnal Blood Metabolome and Effects of Vitamin D Supplementation: A Randomised Crossover Trial in Postmenopausal Women” authors have shown the effect of vitamin D supplementation on the blood metabolome on the diurnal rhythm. This manuscript required a major revision before considering for publication.

Comment 1:   What could be the potential source of acetone metabolism?

Comment 2:  Has acetone studied in metabolism in this type of study? Discuss with appropriate references.

Comment 3: Is that really acetone or other ketone bodies?

Answer comments 1-3: Thank you for these questions. We confirm that we did detect a resonance from acetone with a chemical shift of 2.23 ppm. Acetone is one of the common ketone bodies molecules, which is also comprised by beta-hydroxybutyrate and acetoacetate. Thus, we clarify that what was measured was the metabolite named acetone. Beta-hydroxybutyrate and acetoacetate were not detected and quantified from the obtained NMR spectra. We included a discussion in manuscript.

Changes highlighted in green as follows:

Together with β-hydroxybutyrate and acetoacetate, acetone is a ketone body, and ketone bodies represent a set of fuel molecules serving as an alternative energy source to glucose. Ketone bodies are mainly produced by the liver from fatty acids, and often during periods of fasting, and prolonged or intense physical activity. Ketogenesis is intensified under conditions characterised by insufficient or inaccessible availability of glucose and higher acetone levels are also reported in individuals with diabetes (Dabek et al., 2020). In fact diurnal fluctuations in acetone were initially investigated in connection with establishment of insulin therapy (Åkerblom et Hiekkala, 1970). In the present study, the effect on acetone could indicate an effect of vitamin D on beta-oxidation of fatty acids and thereby the formation and accumulation of ketone bodies.

Comment 4: NMR spectra were recorded not conducted. Correct the sentence.

Answer: We thank for the reviewer collaboration. The sentence was rephrased.

Changes highlighted in green as follows:

NMR spectra were recorded at 310 K on a 600 MHz Bruker Avance III spectrometer equipped with a 5 mm 1H-optimised double resonance broad banded probe (broadband inverse) and fitted with the Bruker SampleJet robot cooling system set to 5 °C.

     Comment 5: How did you confirm pyruvate and acetone in 1D spectrum?

Answer: Thank you for this question. The assignments of pyruvate and acetone were validated by the IVDr tool provided by the Bruker databank, which confirms the metabolite identity in the proton spectrum by J coupling data obtained from a 2D- J resolved spectrum.

Changes highlighted in green as follows:

The obtained NMR spectra were analysed using the In Vitro Diagnostics research (IVDr) tool which for plasma samples discloses 41 metabolites from a database. Of these, 22 plasma metabolites were established to have signal intensities adequate for accurate identification and quantitative evaluation in all the participants' plasma (Figure S1 and S2).

Reviewer 3 Report

In this paper, the authors investigated the acute effect of a single dose of vitamin D3 vs placebo in a population of 29 Caucasian postmenopausal women. They monitored the plasma concentration of 20 metabolites by a metabolomic study performed using NMR spectroscopy.

They confirmed a diurnal rhythm in the metabolome, that is not altered by administration of 200 µg vitamin D3. They found an increase in the concentration of acetone  after the vitamin D administration , suggesting an effect on beta-oxidation.

In my opinion, the manuscript can benefit from some changes:

- They monitored 20 metabolites, but the choice for which these molecules (and not carnitine, choline, or urea etc) should be monitored must be explained in the text.

- in the discussion paragraph, from line 140 to 165, they described other studies that are not comparable with their results. The description of previous studies should be moved from the discussion to the introduction paragraph. In the discussion, only results that can support (or that are conflicting with) their work should be described, i.e. other papers in which the acetone concentration in plasma is related to vit D supplementation.

-Figure S1 shows an example of NMR signal fitting. At least one full spectrum should be attached.

Minor points:

- in the materials and methods they wrote that "30 women gave their consent to participate in the study of whom two dropped out. Twenty-nine participants were included in the present analyses"... If two women dropped out, 28 remain. Do you mean that one dropped out?

- Can you order the number of figures following their appearance in the text?

- Supplementary materials and unpublished results correspond to the same file. On the other hand, there is no citation to unpublished results in the main text

Author Response

REVIEWER 3

Comment: In this paper, the authors investigated the acute effect of a single dose of vitamin D3 vs placebo in a population of 29 Caucasian postmenopausal women. They monitored the plasma concentration of 20 metabolites by a metabolomic study performed using NMR spectroscopy.

They confirmed a diurnal rhythm in the metabolome, that is not altered by administration of 200 µg vitamin D3. They found an increase in the concentration of acetone after the vitamin D administration , suggesting an effect on beta-oxidation.

In my opinion, the manuscript can benefit from some changes:

Comment 1: They monitored 20 metabolites, but the choice for which these molecules (and not carnitine, choline, or urea etc) should be monitored must be explained in the text.

Answer: Thank you for this question. We aimed for an explorative approach and in principle it was untargeted and we thrived to detect as many metabolites as possible. The 20 metabolites reflect the metabolites that we were able to detect and quantify with the robustness and repeatability we had as criteria. We included a discussion in manuscript.

Changes highlighted in green as follows:

The obtained NMR spectra were analysed using the In Vitro Diagnostics research (IVDr) tool which for plasma samples discloses 41 metabolites from a database. Of these, 22 plasma metabolites were established to have signal intensities adequate for accurate identification and quantitative evaluation in all the participants' plasma (Figure S1 and S2).

Figure S2. Example of the 22 metabolites signals fitting of a plasma sample with a p-value match above 95%.  The fitting was obtained using the Bruker IVDr system. The metabolite average of p-value fitting with a match above 95% was used as the cutoff of the list of identified metabolites. The black line, the blue line, and the yellow line represent the original spectrum, the calculated signal fit, and its baseline, respectively. The blue area relates to the metabolite concentration to be determined, the red area represents residues, and the grey area presents the sum of all fitted overlapping signals.

Comment 2: in the discussion paragraph, from line 140 to 165, they described other studies that are not comparable with their results. The description of previous studies should be moved from the discussion to the introduction paragraph. In the discussion, only results that can support (or that are conflicting with) their work should be described, i.e. other papers in which the acetone concentration in plasma is related to vit D supplementation.

Answer: Thank you for this pertinent suggestion, which was promptly attended. The paragraph has been moved to the introduction.

Changes highlighted in green as follows:

However, knowledge about the acute effect of a single high dose of vitamin D3 on the metabolome is sparse. To the best of our knowledge, only one study has investigated the acute effect of a single dose of vitamin D3 on the metabolome. The VITdAL-ICU trial (23) randomised 428 critically ill participants with vitamin D insufficiency to receive either a single dose of 13,500 µg (540,000 IU) vitamin D3 or placebo. In a post-hoc metabolomics study, the authors found significant changes in the metabolome over time relative to the absolute increase in serum 25(OH)D levels from day 0 to day 3. Increases in sphingomyelins, plasmalogens, lysoplasmalogens and lysophospholipids (i.e. metabolites involved in endothelial protection and innate immunity), as well as decreases in phosphatidylethanolamines, amino acid metabolites and acylcarnitines (i.e. metabolites involved in mitochondrial function and fatty acid beta-oxidation) were found. Others have found an association between lipid metabolism and fatty acid oxidation and vitamin D as well. Shirvani et al. (24) reported an increase in metabolites mostly involved in the oxidation of branched-chain fatty acids after 24 weeks with supplementation of either 15 µg, 100 µg or 250 µg vitamin D3 per day, and vitamin D status was associated to lipid metabolism or metabolites of lipid compound origin in the ATBC Cancer Prevention Study (25) and the Hong Kong Osteoporosis Study (26), respectively. However, these metabolic evaluations regard mainly lipids lacking polar metabolites investigations. Recently, it was discovered that a daily supplement of 70 µg (2800 IU) vitamin D3 in 12 weeks increased circulating levels of carnitine, choline, and urea (i.e. muscle-related metabolites negatively correlated with muscle health and physical performance) in postmenopausal women with vitamin D insufficiency (27). Combined with clinical findings reporting negative effects of vitamin D on muscle strength and physical performance (28), this suggested a detrimental effect of moderately high daily doses of vitamin D supplementation on skeletal muscle. However, the daily doses of vitamin D supplementation in the above-mentioned studies were all very high, and in the VITdAL-ICU trial, a mega-dose of 13,500 µg was studied, i.e. multiple times above the 100 µg (4,000 IU) per day which often is considered the upper limit of a safe intake (29). Thus, little is known on the effect of a moderate vitamin D dose on the metabolome. Therefore, the aim of this study was to investigate the diurnal blood metabolome in response to a single high dose of vitamin D3. Our null hypothesis was that the metabolome was unaffected 24 h after vitamin D3 intake, i.e. parallel time-concentration curves and no difference between curves.

Comment 3: Figure S1 shows an example of NMR signal fitting. At least one full spectrum should be attached.

Answer: Thank you for this pertinent suggestion, which were promptly attended. The spectra figure was added as Figure S1.

Changes highlighted in green as follows:

The obtained NMR spectra were analysed using the In Vitro Diagnostics research (IVDr) tool which for plasma samples discloses 41 metabolites from a database. Of these, 22 plasma metabolites were established to have signal intensities adequate for accurate identification and quantitative evaluation in all the participants' plasma (Figure S1 and S2).

Figure S1. ¹H NMR spectrum of a blood plasma sample from a vitamin D supplementated patient (8h after intervention) obtained from A) 1D NOESY presaturation experiment, B) a 1D− Carr−Purcell−Meiboom−Gill (CPMG) spin−echo experiment, and C) a 2D − J-resolved experiment.

Minor points:

Comment 4: in the materials and methods they wrote that "30 women gave their consent to participate in the study of whom two dropped out. Twenty-nine participants were included in the present analyses"... If two women dropped out, 28 remain. Do you mean that one dropped out?

Answer: Thank you for this pertinent suggestion, which were promptly attended, as follows:

Changes highlighted in green as follows:

Participants were recruited from the general background population by direct mailing using a list of randomly selected individuals living in the area of Aarhus, Denmark. In total, 8,977 invitations were sent out. Of 774 respondents, 243 were eligi-ble for a blood test. Seventy-one had vitamin D insufficiency. Among these, 30 gave their consent to participate in the study of whom one dropped out. Twenty-nine participants were included in the present analyses (Figure 3).

Comment 5: Can you order the number of figures following their appearance in the text?

Answer: Thank you for this pertinent suggestion, which was promptly attended.

Comment 6: Supplementary materials and unpublished results correspond to the same file. On the other hand, there is no citation to unpublished results in the main text.

Answer: Thank you for this pertinent suggestion. This was a mistake as there are no unpublished results.

Round 2

Reviewer 2 Report

 Comment 5: How did you confirm pyruvate and acetone in 1D spectrum?

Answer: Thank you for this question. The assignments of pyruvate and acetone were validated by the IVDr tool provided by the Bruker databank, which confirms the metabolite identity in the proton spectrum by J coupling data obtained from a 2D- J resolved spectrum

What did author actually see in J-resolved, since both pyruvate and acetone show singlet? Was it C13 J-Resolved or 1H J-Resolved experiment?

Author Response

Thank you for your pertinent question. We apologise for the general answer. The assignments of pyruvate and acetone were validated by the HSQC spectrum. For this, we added a new Figure in Support Information.

Furthermore, we have added the lines:

"1H-13C HSQC experiments were performed on selected samples."  

and

"IVDr metabolites identity was confirmed by HSQC data."

to section 4.3 in the manuscript.